# Burden of Family Caregivers of Patients with Oral Cancer in Home Care in Taiwan

**DOI:** 10.3390/healthcare11081107

**Published:** 2023-04-12

**Authors:** Tzu-Ting Chang, Shu-Yuan Liang, John Rosenberg

**Affiliations:** 1Department of Nursing, Taipei Veterans General Hospital, Taipei 112, Taiwan; 2School of Nursing, National Taipei University of Nursing and Health Sciences, Taipei 112, Taiwan; 3School of Health, University of the Sunshine Coast, Caboolture, QLD 4059, Australia

**Keywords:** burden, family caregiver, oral cancer

## Abstract

Oral cancer is currently the fourth leading cause of cancer-related death in Taiwan. The complications and side effects of oral cancer treatment cause a tremendous burden on patients’ family caregivers. This study explored the burden on primary family caregivers of patients with oral cancer and its related factors. One hundred and seven patients with oral cancer and their primary family caregivers were included through convenience sampling. The Caregiver Reaction Assessment (CRA) scale was employed as the primary research instrument. The primary factors of caregiver burden, in descending order, were disrupted schedules (M = 3.19, SD = 0.84), a lack of family support (M = 2.82, SD = 0.85), health problems (M = 2.67, SD = 0.68), and financial problems (M = 2.59, SD = 0.84). The CRA scores of the caregivers differed significantly in terms of education level (t = 2.57, *p* < 0.05) and household income (F = 4.62, *p* < 0.05), which significantly predicted caregiver burden (R2 = 0.11, F = 4.32, *p* = 0.007). The study results provide a reference for healthcare professionals to identify the factors for family caregiver burden, as well as the characteristics of patients and family caregivers particularly vulnerable to caregiver burden, thus improving family-centred care.

## 1. Introduction

In Taiwan, oral cancer is associated with alcohol and betel nut consumption [1,2]. More than 90% of the patients who have died of oral cancer in Taiwan have been men; oral cancer is the fourth most prominent cause of cancer-related death among men [3]. The complications, side effects, and risks of death associated with oral cancer treatment present a tremendous burden to the patients, patient families, and medical professionals in care provision.

Although oral cancer can be treated through a combination of surgery, chemotherapy, and radiotherapy, surgery is the main treatment of choice [4]. However, surgery alters the tissue structures of patients’ mouths, faces, and jawbones, thus altering their facial appearances, chewing and swallowing functions, and communication abilities, in addition to causing pain [5,6,7,8]. Furthermore, patients are burdened by the side effects of chemotherapy and radiotherapy. Therefore, challenges in caregiving to patients with oral cancer can be greater than those in caregiving to patients with other types of cancer.

Families play a major role in caregiving to patients with oral cancer in Taiwan; families are required to provide direct home care and economic, social, and emotional support [9]. This challenging caregiving process causes a great burden on the time, finances, and health of these families [9]. It also causes fatigue and emotional effects on the caregiving families, severely affecting their physical and psychological health in the process [10]. Consequently, this can reduce the effectiveness of families’ care for patients with oral cancer [11]. Studies report not all family caregivers of patients with oral cancer have negative experiences. Some family caregivers report positive experiences in caregiving to patients with oral cancer in terms of discovering the importance of caregiving for their family members with oral cancer and feeling an enhanced sense of dignity [12,13,14,15]. Some factors, however, such as the sociodemographic variables of patients and their families and the patient’s medical condition, may exacerbate the caregiver burden.

Studies indicate a correlation between caregiver burden and age, sex, education level, marital status, religious affiliation, employment status, financial status, relationship with patients, whether they live with patients, length of care for patients, and whether they are supported by other family members [15,16,17,18,19,20]. Studies also report the possibility of a correlation between caregiver burden and patients’ age, education level, religious affiliation, history of chronic diseases along with the disease stages, and type of treatment [15,17,18,19].

The study of the burden of primary family caregivers to patients with oral cancer and its related factors will help medical professionals to more clearly understand the caregiver burden status and, accordingly, provide family-centred care. The present study explored the burden of primary family caregivers in providing home care to patients with oral cancer and its related factors.

## 2. Methods

### 2.1. Study Design, Sample, and Procedure

A cross-sectional descriptive study was conducted, with convenience sampling. A structured questionnaire was administered from May 2016 to May 2018 to 107 outpatients with oral cancer from the radiology department in a teaching hospital in Northern Taiwan and their primary family caregivers. The inclusion criteria for the patients were (1) age 20 years or older, (2) diagnosis of oral cancer, and (3) receipt of surgery, chemotherapy, or radiotherapy targeting oral cancer. The inclusion criteria for the patients’ primary family caregivers were as follows: (1) age 20 years or older, (2) primary family caregivers as recognised by the patients, and (3) living together with the patients. The exclusion criteria for primary caregiver were the primary caregiver had an employment relationship with the patient. This study was approved by the institutional review board of the hospital, and informed consent forms were signed by the patients and their primary family caregivers. The questionnaires were distributed by the research assistants and voluntarily filled in by the caregivers, after which the research assistants examined the responses on-site to check for any unanswered items. The family caregivers were requested by the assistants to fill in any items that were not answered. The patients’ medical characteristics were collected from their medical history by the research assistants.

### 2.2. Ethical Considerations

This study was approved by the institutional review board of a teaching hospital (VGHIRB no.: 2014-04-001AC) in Northern Taiwan. The research assistants verbally explained the research objective, data protection method, and research procedure to the participants before acquiring signed consent from the participants. The participants’ personal information was coded in the questionnaire to protect their privacy. Participants who were unwilling to continue being surveyed or unfit for further survey because of poor physical condition were free to withdraw from the study, and their data collection was discontinued by the researchers.

### 2.3. Measures

#### Sociodemographic Variables

The data collected in this study were the sociodemographic variables of the patients and their primary family caregivers, as well as the types of care and the patients’ medical characteristics. The sociodemographic variables included sex, age, marital status, education level, religious affiliation, employment status, and household income. The medical characteristics included the time of oral cancer diagnosis, the stage of cancer, current treatment status, and reported side effects from the treatment. The data on the primary family caregivers also included their relationship with the patients and the type and length of care.

### 2.4. Caregiver Reaction Assessment

Caregiver Reaction Assessment (CRA), a tool used to assess caregiver burden, consists of 5 subscales encompassing 24 items, namely, disrupted schedules (5 items), financial problems (3 items), health problems (4 items), lack of family support (5 items), and self-esteem (7 items). While disrupted schedules, financial problems, health problems, and lack of family support constituted negative caregiver reactions, self-esteem constituted positive caregiver reactions. A 5-point Likert scale was employed (1 = strongly disagree; 5 = strongly agree), with a higher score indicating a higher caregiver burden [21]. The Cronbach’s α of each subscale ranged between 0.68 and 0.90 [22,23]. According to the test–retest reliability test results, the intraclass correlation of the Chinese language edition of the scale was ≥0.75, and the edition exhibited sufficient construct validity [24]. The Cronbach’s α of each subscale in this edition ranged between 0.70 and 0.92, and the total Cronbach’s α of the edition was 0.82 [20].

### 2.5. Statistical Analysis

SPSS 22.0 for Windows was employed for statistical data analysis. Descriptive statistical analysis was performed on the means, standard deviations, frequencies, and percentages for the sociodemographic variables of all the participants, the patients’ medical characteristics, the relationship between the primary family caregivers and the patients, the type and length of care, and caregiver burden (according to the CRA). The differences in caregiver burden based on various sociodemographic variables and medical characteristics were examined using analysis of variance and an independent sample *t*-test. The correlation between the CRA scores and the caregivers’ and patients’ age, length of patient care, and length of patient illness was analysed through Pearson product–moment correlation. Subsequently, multiple regression analysis was conducted on the caregiver burden predictive power of the sociodemographic variables and medical characteristics.

When certain sociodemographic variables and medical characteristics were significantly correlated with caregiver burden, dummy coding was to be conducted for the variables and characteristics that are discrete or nominal before conducting multiple regression analysis. During the multiple regression analysis, the sociodemographic variables, medical characteristics, the relationships between the primary family caregivers of the patients, and the types and lengths of care that are significantly correlated with caregiver burden were treated as independent predictors of caregiver burden, which is the dependent variable.

## 3. Results

### 3.1. Sociodemographic Variables of the Primary Family Caregivers

Of the 107 primary family caregivers, 98 were women (91.6%). The average age of these caregivers was 51 ± 10.8 years, and their ages ranged from 20 to 70 years. Regarding the relationships between the caregivers and patients, 78 of the caregivers were spouses to the patients (72.9%). Of these caregivers, 60 had senior high school or higher levels of education (56.1%); 94 were married (87.9%); 28 were employed and provided care to their patients after work (26.2%); 51 had household incomes of no higher than NT$500,000 (47.7%); 93 had religious affiliations (86.9%); and 28 were diagnosed with chronic diseases themselves (26.2%; Table 1).

### 3.2. Types of Care Provided by the Primary Family Caregivers

Of the primary family caregivers, 44 provided care to their patients together with other family members or hired caregivers (41.1%), whereas 28 had to care for their patients independently throughout the day (26.2%). The primary family caregivers had provided care to their patients for an average of 36.4 ± 40.3 months, with the length of care ranging from 1 to 171 months. Of the caregivers, 43 had provided full care for patients each week (40.2%); 40 had irregular rest time each week (37.4%); and 89 were inexperienced in providing care to patients (83.20%; Table 1).

### 3.3. Sociodemographic Variables of the Patients

Of the 107 patients with oral cancer, 100 were men (93.5%), and the average age was 56.4 ± 9.7 years (range, 33–89 years). Of the total patients, 59 graduated from educational institutes no higher than junior high schools (55.1%); 38 remained employed in full-time jobs even after developing oral cancer (35.5%), but 37 were unemployed (34.6%); 88 lived with their families or friends (82.2%); and 87 had a religious affiliation (87.3%; Table 2).

### 3.4. Medical Characteristics

The patients had experienced oral cancer for 1–171 months, with an average of 42.5 ± 44.4 months. Of all the patients, 39 were at the fourth stage of oral cancer (36.4%), and 32 were at the second stage (29.9%); 84 had completed their treatment (78.5%); and 39 experienced the side effects of their treatment (36.4%; Table 2).

### 3.5. Caregiver Burden

CRA was used to assess the caregiver burden. Specifically, the assessment subscales were disrupted schedules, financial problems, health problems, a lack of family support, and self-esteem. The results reveal the highest average scores in disrupted schedules (3.19 ± 0.84), followed by a lack of family support (2.82 ± 0.85), health problems (2.67 ± 0.68), and financial problems (2.59 ± 0.84) among primary family caregivers to patients with oral cancer. However, the caregivers’ feelings about providing care to their family members were not always negative. The caregivers obtained an average score of 3.58 ± 0.49 in self-esteem (Table 3). Thereafter, the self-esteem was scored in reverse, and the total average caregiver burden score was calculated; a higher total average score indicates a higher caregiver burden. The total average score was calculated as 2.74 ± 0.52, with the total score of each caregiver ranging from 1.31 to 3.87.

Higher scores indicate a higher burden, with the exception of scores in self-esteem, where a higher score indicates a lower burden.

### 3.6. Differences in the Burden of the Primary Family Caregivers According to Their Sociodemographic Variables and Type of Care

After self-esteem was scored in reverse, the total average score of the burden of the primary family caregivers was calculated; a higher score indicates a higher caregiver burden. The association of the sociodemographic variables and type of care with total burden was then investigated. The age of the caregivers (*r* = 0.12, *p* > 0.05) and the length of care (*r* = 0.07, *p* > 0.05) were not significantly correlated with the total average burden. Only the education level (*t* = 2.57, *p* < 0.05) and the household income (*F* = 4.62, *p* < 0.05) of the caregivers were significantly correlated with the total average burden, but no other sociodemographic variables and no type of care were significantly correlated with the total burden (Table 1).

### 3.7. Differences in Caregiver Burden According to Patients’ Sociodemographic Variables and Medical Characteristics

The age of the patients (*r* = 0.04, *p* > 0.05) and the length of illness (*r* = 0.09, *p* > 0.05) were not significantly correlated with caregiver burden. Furthermore, the other sociodemographic variables and medical characteristics were not significantly correlated with the caregiver burden (Table 2).

### 3.8. Predictive Power of Sociodemographic Variables, Medical Characteristics, and Type of Care on Caregiver Burden

The education level and household income of the primary family caregivers were significantly correlated with the total burden. These variables were further analysed for their power of predicting caregiver burden through multiple regression. The collinearity tolerance values ranged between 0.44 and 0.92 (higher than the cut-off value of 0.10), and the variance inflation factors ranged between 1.09 and 2.26 (lower than the cut-off value of 10), indicating that the variables exhibited no collinearity [25]. In the multiple regression analysis, the education level and household income of the primary family caregivers were analysed through the enter approach and selected as the predictor variables for the total caregiver burden. These predictor variables significantly predicted the total variance of the caregivers’ burden (11%; *p* = 0.007) (Table 4).

## 4. Discussion

This study examined the burden on primary family caregivers to patients with oral cancer and its related factors. The results may help medical professionals to understand the current status regarding caregiver burden and identify the family and patient characteristics that incur caregiver burden, facilitating family-centred care. The results of this study reveal that the primary family caregivers exhibited high self-esteem; disrupted schedules represented the most prominent factor to their burden, followed by a lack of family support, health problems, and financial problems.

Disrupted schedules were reported by the primary family caregivers as the greatest contributing factor to their burden, which is consistent with the findings of studies on the burden on primary family caregivers to patients with rectal, lung, oral, and terminal cancer [9,12,13,14,15,17,18,20]. In this study, nearly all the primary family caregivers were wives of the patients with oral cancer. Based on the findings regarding employment status and age, most caregivers were required to manage their jobs and take care of their young children in addition to providing care to the patients.

A lack of family support constituted the second most prominent factor to caregiver burden. This differs from the findings of most studies, which have indicated that a lack of family support is the least prominent contributor to caregiver burden [9,12,13,14,15,17,18,20]. In this study, only one-fourth of the caregivers provided care independently throughout the day; most of the caregivers provided care with the help of other people or provided care independently but not throughout the day. However, the helpers may not be members of the caregivers’ families but caregivers hired from other countries, which may have caused the primary family caregivers to perceive family support as lacking. Supportive communication between family members is critical [26].

Health problems were the third highest contributor to caregiver burden, which is consistent with the findings in some of the existing studies [9,12,13,17]. Approximately one-fourth of the caregivers were themselves diagnosed as having chronic diseases, and most of the caregivers had received help from other people in providing care. Although health problems were only the third highest factor for caregiver burden, family caregivers experienced emotional stress in addition to physical fatigue in providing care to patients; this affects the caregivers’ overall psychological health [27].

Many studies have indicated financial problems as the second greatest contributor to caregiver burden [9,12,13,17], but this study revealed it as the least prominent factor. In Taiwan, national health insurance covers the health care costs of the population in Taiwan, and patients with cancer are provided additional subsidies related to major diseases. Therefore, problems related to medical expenses may not be the primary factor for the burden of primary family caregivers. Nevertheless, the financial conditions of the caregivers’ families contribute to the caregiver burden. Specifically, the caregivers with the lowest household incomes exhibited significantly heavier total burdens than the other caregivers.

In fact, household incomes significantly predict the overall burden of primary family caregivers. According to Cheng et al. [27], family caregivers of patients with oral cancer are required to deal with the treatment side effects. For example, patients may require nasogastric tube feeding or emergency treatment, which involves additional expenses. Occasionally, an additional cost is required to hire replacement caregivers. In Taiwan, oral cancer is primarily associated with betelnut consumption [1,2], and most betelnut consumers are working-class people with disadvantaged household economic conditions. Moreover, families may sometimes be required to resign from their jobs to take care of patients, further worsening their financial conditions. According to Cheng et al. [27], acquiring financial support is a critical problem faced by primary family caregivers. Therefore, financial problems remain a critical problem to these caregivers. Patients in Taiwan are included in the national health insurance system, so the financial problem is not the primary burden in the current population. However, according to the statistical results of the current study, financial status is a crucial variable that can significantly predict the burden of a family. Financial problems may still play an important role in the family burden for other populations not covered by health insurance.

The primary family caregivers scored high in self-esteem. Because self-esteem is a positive aspect of caregiver reactions, a higher score in self-esteem indicates a lower caregiver burden. This is consistent with the findings of most studies [9,12,13,14,15], indicating that the caregivers were willing to care for their family members who were ill and considered the task critical despite the huge burden it caused.

After self-esteem was scored in reverse, the total average caregiver burden score was calculated. A higher score indicates a higher total caregiver burden. The results of this study reveal that the education levels and household incomes of the primary family caregivers significantly predicted their total burden. The caregivers with higher education levels exhibited lower burdens than those with lower education levels, whereas those with lower household incomes experienced higher burdens than those with higher household incomes. Studies have reported that financial and educational problems in a family critically affect the caregiver burden [20,28,29]. Accordingly, medical professionals should pay utmost attention to family caregivers with low incomes or education levels and understand their needs in providing care.

A meta-analysis has shown that case management, psychoeducation, and multicomponent interventions can significantly reduce the burden on caregivers. In particular, case management and counselling appeared to be better than cognitive behavioural therapy [30]. The multiple components of REACH II intervention focused on social support, communication, selfcare, emotional well-being, and community support [31,32,33], which included access to support groups by videophone [33,34,35]. A case management program is specifically their adaptability and flexibility, which provides caregivers with the ability to respond to the complex needs of the family member they care for [36].

This study was a cross-sectional descriptive study and did not clarify the changes in the burden of primary family caregivers over changes in patients’ conditions or time. Moreover, because participants were enrolled from only one teaching hospital in Northern Taiwan, the results may not be representative of all primary family caregivers. Convenience sampling used in the current study may cause sampling bias. Families with high care burdens may be eliminated inherently. On the other hand, the sample size was small for several sociodemographic and medical variable groups. It is unlikely that statistical differences could be detected in caregiver burden by patients’ sociodemographic information and medical characteristics.

## 5. Conclusions

The primary factors contributing to the burden of primary family caregivers, in descending order, were disrupted schedules, a lack of family support, health problems, and financial problems. The results of this study reveal that the caregivers exhibited high self-esteem, which is a positive aspect of caregiver reactions. Although home care presented a huge burden to the primary family caregivers, the caregivers were still willing to provide care and considered it pivotal. Moreover, low household income and low education levels significantly affected the caregiver burden.

Medical professionals should prioritise the arrangement of primary family caregivers’ time for care in their education strategies. Referral of related care resources is also crucial in providing family caregivers adequate time to rest. Additionally, supportive communication between family members must be promoted. Self-care strategies should be taught to family caregivers. Furthermore, economically disadvantaged families, particularly those with low education levels, should be assisted in finding substantial support from social welfare institutions.

This study recommends that future research builds on the results of this current study and focuses on the development of relevant interventions to reduce the burden of primary caregivers at home.

## Figures and Tables

**Table 1 healthcare-11-01107-t001:** Differences in total burden of the primary family caregivers according to their sociodemographic variables and the type of care (*n* = 107).

Variable	Number	%	Mean	SD	*t/F*	*p*
Sex					*t* = 0.83	0.41
Male	9	8.4	2.87	0.29		
Female	98	91.6	2.72	0.53		
Relationship with patient					*t* = 1.19	0.24
Spouse	78	72.9	2.77	0.48		
Other	29	27.1	2.64	0.60		
Education level					*t* = 2.57 *	0.02
Junior high school or lower	47	43.9	2.87	0.40		
Senior high school or higher	60	56.1	2.63	0.57		
Marital status					*t* = 1.88	0.06
Married/cohabiting	94	87.9	2.77	0.49		
Other	13	12.1	2.49	0.63		
Employment status					*F* = 0.62	0.60
Quit their job	19	17.8	2.86	0.49		
Employed and providing care after work	28	26.2	2.65	0.55		
Employed but on leave	11	10.3	2.78	0.43		
Other	49	45.8	2.73	0.53		
Household income					*F* = 4.62 *	0.01
➀ ≤NT$500,000	51	47.7	2.89	0.49	➀ > ➁	
➁ NT$510,000–NT$1,000,000	39	36.4	2.59	0.52		
➂ ≥NT$1,010,000	17	15.9	2.62	0.47		
Religious affiliation					*t* = 0.71	0.49
Yes	93	86.9	2.76	0.48		
No	14	13.1	2.62	0.71		
Chronic disease					*t* = 0.07	0.94
Yes	28	26.2	2.73	0.43		
No	79	73.8	2.74	0.54		
Type of care						
Together with someone else	44	41.1	2.68	0.52	*F* = 2.75	0.07
Independent care throughout the day	28	26.2	2.93	0.36		
Independent care without provision throughout the day	35	32.7	2.66	0.58		
Length of care per week					*F* = 1.79	0.17
No rest	43	40.2	2.85	0.39		
Rest each week	24	22.4	2.63	0.51		
Irregular rest	40	37.4	2.68	0.61		
Patient care experience						
Yes	18	16.8	2.87	0.63	*t* = −1.17	0.25
No	89	83.2	2.71	0.49		

* *p* < 0.05. Total burden is calculated with self-esteem scored in reverse; a higher score indicates higher burden.

**Table 2 healthcare-11-01107-t002:** Differences in total burden according to patients’ sociodemographic information and medical characteristics (*n* = 107).

Variable	Number	%	Mean	SD	*t/F*	*p*
Sex					*t* = 1.43	1.56
Male	100	93.5	2.76	0.51		
Female	7	6.5	2.47	0.50		
Education level					*t* = −1.23	0.22
Junior high school or lower	59	55.1	2.68	0.61		
Senior high school or higher	48	44.9	2.80	0.36		
Employment status					*F* = 1.30	0.28
Unemployed	37	34.5	2.87	0.54		
Full-time	38	35.5	2.67	0.50		
Retired	16	15.0	2.70	0.44		
Other	16	15	2.62	0.54		
Living with family or friends					*t* = 0.50	0.62
Yes	88	82.2	2.75	0.54		
No	19	17.8	2.68	0.39		
Religious affiliation					*t* = −0.29	0.78
Yes	87	87.3	2.74	0.48		
No	20	18.7	2.71	0.64		
Stage of cancer					*F* = 0.97	0.41
Stage 1	22	20.6	2.62	0.52		
Stage 2	32	29.9	2.71	0.62		
Stage 3	14	13.1	2.70	0.50		
Stage 4	39	36.4	2.84	0.41		
Received treatment					*t* = 0.54	0.59
Yes	23	21.5	2.69	0.63		
No	84	78.5	2.75	0.48		
Side effects from treatment					*t* = 0.41	0.68
Yes	39	36.4	2.71	0.54		
No	68	63.6	2.75	0.50		

Total burden is calculated with self-esteem scored in reverse; a higher score indicates higher burden.

**Table 3 healthcare-11-01107-t003:** Caregiver burden (*n* = 107).

Variable	Mean	SD	Minimum	Maximum
Disrupted schedule	3.19	0.84	1.20	5.00
Financial problem	2.59	0.84	1.00	4.60
Lack of family support	2.82	0.85	1.00	5.00
Health problem	2.67	0.68	1.00	4.50
Self-esteem	3.58	0.49	2.43	4.57

**Table 4 healthcare-11-01107-t004:** Multiple Regression Analysis for Variables Predicting Family Caregiver Burden (*n* = 107).

Variable	*B*	SE *B*	*β*
Caregivers’ education levels			
Senior high school or higher vs. junior high school or lower	−0.19	0.10	−0.18
Household income/year (NTD)			
≤NT$500,000 vs. ≥NT$1,010,000	0.20	0.14	0.19
NT$510,000–NT$1,000,000 vs. ≥NT$1,010,000	−0.07	0.15	−0.07
Overall model	*R^2^ =* 0.11 (*F* (3, 103) *=* 4.32, *p* = 0.007)		

NTD—New Taiwan Dollar.

## Data Availability

The data presented in this study are available from the corresponding author upon reasonable request.

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
