# Peer review of "Burden of Family Caregivers of Patients with Oral Cancer in Home Care in Taiwan"

_healthcare, 2023, doi:10.3390/healthcare11081107_

Round 1

Reviewer 1 Report

The study examined the burden of primary family caregivers to patients with oral

cancer and its related factors.

The study is highly innovative, but there are still some deficiencies in conclusion verification.

1. The format of Table 1and 2 needs to be more standardized to facilitate readers' reading.

2. What were the exclusion criteria for study subjects?

3. Why was the patient's condition and treatment outcome not significantly associated with caregiver burden? This question requires a detailed explanation by the authors.

Author Response

Response to Reviewer 1 Comments

Comments and Suggestions for Authors

The study examined the burden of primary family caregivers to patients with oral cancer and its related factors.

The study is highly innovative, but there are still some deficiencies in conclusion verification.

  1. The format of Table 1and 2 needs to be more standardized to facilitate readers' reading.

Response: In the tables 1-2 we indicate either t or F before the values (eg. t =0.83, t =1.19, t =2.57…). As well, we add p value in the tables 1-2 to show no significant or significant differences in family burden.

  1. What were the exclusion criteria for study subjects?

Response: We add “The exclusion criteria for primary caregiver was the primary caregiver who had an employment relationship with the patient.”

  1. Why was the patient's condition and treatment outcome not significantly associated with caregiver burden? This question requires a detailed explanation by the authors.

Response: We add “The sample size was small for several of sociodemographic and medical variable groups. It is unlikely that statistical differences could be detected in caregiver burden by patients’ sociodemographic information and medical characteristics.” in the research limitation section.

Reviewer 2 Report

Dear authors. It has been very gratifying for me to be able to review your work. I found the topic very interesting and necessary. In the work they have analyzed the burden of caregivers and its related factors.

I would like to send you some comments/reflections:

- I consider that methodologically the work is correct.

- The bibliography is current and appropriate to the subject of study.

- I invite you to reflect and include limitations in the study since they are not included. The study has been carried out in a population where all people have access to the national health system, hence the financial issue is not relevant in this population. What would they have to improve if this work is extrapolated to other populations?

- I think they should include what can be done to avoid overloading family carers. Where, at what time and when should all available resources be put into operation so that a caregiver does not suffer from this overload? Very interesting future lines of research are opened.

Thank you

Author Response

Response to Reviewer 2 Comments

Comments and Suggestions for Authors

Dear authors. It has been very gratifying for me to be able to review your work. I found the topic very interesting and necessary. In the work they have analyzed the burden of caregivers and its related factors.

I would like to send you some comments/reflections:

- I consider that methodologically the work is correct.

- The bibliography is current and appropriate to the subject of study.

- I invite you to reflect and include limitations in the study since they are not included. The study has been carried out in a population where all people have access to the national health system, hence the financial issue is not relevant in this population. What would they have to improve if this work is extrapolated to other populations?

Response: We add “Patients in Taiwan are included in the national health insurance system, so the financial problem is not the primary burden in the current population. However, according to the statistical results of the current study, the financial status is a crucial variable that can significantly predict the burden of family. Financial problem may still play an important role for the family burden for other population is not covered by health insurance.In addition, in the research limitation section, we add “Convenience sampling used in the current study may cause sampling bias. Families with high care burdens may be eliminated inherently. On the other hand, the sample size was small for several of sociodemographic and medical variable groups. It is unlikely that statistical differences could be detected in caregiver burden by patients’ sociodemographic information and medical characteristics..

- I think they should include what can be done to avoid overloading family carers. Where, at what time and when should all available resources be put into operation so that a caregiver does not suffer from this overload? Very interesting future lines of research are opened.

Response: We add “A meta-analysis has shown that case management, psychoeducation and multicomponent interventions can significantly reduce the burden of caregivers. In particular, case management and counseling appeared to be better than cognitive behavioral therapy [30]. The multiple components of REACH II intervention focused on social support, communication, selfcare, emotional well-being, and community support [31-33], which included access to support groups by videophone [33-35]. A case management program is specifically their adaptability and flexibility, which provides caregivers with the ability to response the complex needs of family member they care for [36].” In addition, in conclusion section we add “This study recommends that future researches build on the results of this current study and focus on the development of relevant intervention to reduce the burden of primary caregivers at home.”